# Cell-Free Therapies: The Use of Cell Extracts to Mitigate Irradiation-Injured Salivary Glands

**DOI:** 10.3390/biology12020305

**Published:** 2023-02-14

**Authors:** Xinyun Su, Akshaya Upadhyay, Simon D. Tran, Zhengmei Lin

**Affiliations:** 1Guangdong Provincial Key Laboratory of Stomatology, Department of Operative Dentistry and Endodontics, Guanghua School of Stomatology, Sun Yat-sen University, Guangzhou 510055, China; 2McGill Craniofacial Tissue Engineering and Stem Cells Laboratory, Faculty of Dental Medicine and Oral Health Sciences, McGill University, Montreal, QC H2X 2J7, Canada

**Keywords:** conditioned media, mesenchymal stem cells, radiation damage, regeneration, regenerative medicine, salivary gland, stem cell extracts, stem cell therapy, xerostomia

## Abstract

**Simple Summary:**

Radiation therapy for cancer treatment can lead to damage to surrounding healthy tissues and organs, like salivary glands. Loss of salivary function leads to a decrease in the quality and quantity of saliva, which increases the chances of oral and systemic infections; causes difficulty in swallowing and loss of taste leading to malnutrition. Current treatment modalities to counter this effect are unable to bring satisfaction to the patients whose quality of life is considerably reduced due to cancer and radiotherapy. Here we provide a comprehensive overview of prospective treatment strategies using stem cell-derived extracts and their comparison with other cell-free- therapies. This work is to serve as a resource for scientists to understand, apply and accelerate this treatment methodology in the clinics where its role will be vital to face this clinical challenge.

**Abstract:**

Radiotherapy is a standard treatment for head and neck cancer patients worldwide. However, millions of patients who received radiotherapy consequently suffer from xerostomia because of irreversible damage to salivary glands (SGs) caused by irradiation (IR). Current treatments for IR-induced SG hypofunction only provide temporary symptom alleviation but do not repair the damaged SG, thus resulting in limited treatment efficacy. Therefore, there has recently been a growing interest in regenerative treatments, such as cell-free therapies. This review aims to summarize cell-free therapies for IR-induced SG, with a particular emphasis on utilizing diverse cell extract (CE) administrations. Cell extract is a group of heterogeneous mixtures containing multifunctional inter-cellular molecules. This review discusses the current knowledge of CE’s components and efficacy. We propose optimal approaches to improve cell extract treatment from multiple perspectives (e.g., delivery routes, preparation methods, and other details regarding CE administration). In addition, the advantages and limitations of CE treatment are systematically discussed by comparing it to other cell-free (such as conditioned media and exosomes) and cell-based therapies. Although a comprehensive identification of the bioactive factors within CEs and their mechanisms of action have yet to be fully understood, we propose cell extract therapy as an effective, practical, user-friendly, and safe option to conventional therapies in IR-induced SG.

## 1. Introduction

Salivary glands (SG) hypofunction is one of the most common side effects of irradiation (IR) in head and neck cancer (HNC) patients. Up to 80% of these patients who received radiotherapy (RT) consequently suffer from xerostomia (which defines subjective dry mouth symptoms) and hyposalivation (an objective decrease in salivary flow rate) because of the irreversible damage to acinar cells in SGs caused by IR [1,2,3]. The functional cells in SGs are highly differentiated and relatively quiescent, but they are surprisingly radiosensitive, especially the acinar cells [4]. Thus, a rapid decrease in salivary secretion occurs in the first-week post-RT, and a further significant decrease is observed during the three months following RT [2]. In addition to volume, other factors, such as salivary electrolyte levels, buffering properties, and saliva’s antibacterial systems, are also changed post-radiotherapy [1,2]. The pH of the saliva is reduced from 7.0 to 5.0 [2,5,6], and the concentration of inorganic salt (such as sodium, chloride, and calcium) and organics (such as immunoproteins and lysozyme) are increased. RT-induced saliva reduction is commonly associated with the increased activity of tooth caries, oral fungal infections, and mucosal inflammation [7]. Such related symptoms can consequently affect the individual’s nutritional intake, speech, and sleep, thus reducing their quality of life.

The process of IR-induced SG injury has been divided into four stages, based on the time phases in the rat model [8]. In phase one (0–10 days), water secretion is affected with no apparent cell death, while in phase two (10–60 days) and phase three (60–120 days), the amylase secretion significantly decreases, and acinar cells disappear. Then, a lower salivary flow rate is observed in phase four (120–240 days), with loss of functional cells and their supportive environment (of ductal and stem cells). Recently, another study further summarized the classification of IR-induced SG damage into two stages: the acute (0–3 days post-IR) and chronic phases (>30 days post-IR) in mouse and rat models [4]. Both classifications explored relative histopathological changes and the related mechanism of IR-induced SG injury. During the acute stage, the secretory function decreases, with no severe apoptosis or cell loss observed in SGs [8,9,10]. Acute SG dysfunction is partially attributed to compromised M3-muscarinic receptors and water channels (e.g., aquaporin-5, AQP5) in the plasma membrane during and after the radiation [11]. At the delayed stage, the amylase secretion, the saliva flow rate, and the volume of irradiated salivary glands were significantly decreased [12]. Irradiation can damage the cellular DNA via free radicals in the nucleus and mitochondria and induce the death of the reproductive and functional cells (e.g., acinar progenitor cells and endothelial progenitor cells) in SGs [13,14,15]. From 30 to 300 days post-IR, fibrosis, and fatty degeneration develop and consequently increase the dysfunction of salivary glands [4,12]. These mechanisms could also be an interpretation of the later cell loss, cell apoptosis, salivary secretion, and blood flow reduction during the chronic phase [16]. To summarize, irradiation-induced SG damage can be divided into two mechanisms: (1) the acute phase, i.e., cellular dysfunction due to cell membrane damage, and the (2) delayed phase, i.e., a classical killing of progenitor cells because of DNA damage and disorders of the cellular microenvironment [4,8].

Most current treatment options for IR-induced SG hypofunction are palliative [17] or preventive, resulting in a limited efficacy [18,19,20]. The main objective of palliative therapy, such as salivary substitutes, is to relieve the symptoms and reduce the discomfort in patients with xerostomia but not to stimulate the salivary gland to secrete natural saliva [21]. In other words, these treatments do not aim to reverse the acute damage to the cells or protect the progenitor and functional cells in SG. Therefore, the effect of these salivary substitutes is transient [21,22]. As another palliative therapy, the sialagogue treatment can stimulate the secretory function of SGs; however, it becomes ineffective if SG cells are already damaged and insufficient before the treatment.

Additionally, it has severe side effects and inconsistent outcomes for certain patients, especially those of advanced age [23]. As for preventive therapies, such as intensity-modulated radiotherapy (IMRT) and radical scavengers, they have uncertain effectiveness in clinical trials [24,25,26,27]. Furthermore, surgical treatment (e.g., salivary gland transfer) is underutilized due to its invasiveness and complications, such as ipsilateral facial edema and neck numbness [28,29,30]. Therefore, regenerative therapy has garnered tremendous interest in the field of treatment for SG hypofunction. Current knowledge of restoring saliva secretion comes from various experimental strategies, such as cell-based therapies [31,32,33] and cell-free therapies [34,35]. Cell-based therapies have been reported as a regenerative option to increase the salivary flow rate and relieve the xerostomia caused by radiotherapy [36,37]. Various cells, including stem cells and non-stem cells from different tissues, restored IR-induced SG dysfunction by reducing cell apoptosis and protecting the structure and function of acinar cells in SGs [37,38,39]. Recently, stem cell treatments have been investigated in phase I–II clinical trials [33,40,41]. Evidence suggests that the paracrine effect is the primary mechanism for cell-based therapy in the SG post-IR [37,38,42]. The paracrine factors released from cells allow tissue repair and regeneration via modulating the immune reaction, mitigating inflammation and fibrotic effects, promoting angiogenesis and neurogenesis, and preventing apoptosis [37,43,44,45]. Based on this theory, various cell-free therapies have been developed in the past few decades, including cell extract therapy, conditioned medium therapy, and others (e.g., extracellular vesicle therapy). This review focuses on cell-free therapy as an alternate strategy to palliative or conventional preventive treatments, with a particular emphasis on using cell extract (CE). First, we summarize a variety of cell extracts and their therapeutic uses in SG. Then, we provide optimization approaches for cell extract treatments from different aspects and compare CE with several other cell-free derivatives for the treatment of IR-induced SG hypofunction.

## 2. Cell Extract Therapies and IR-Damage SG

The cell extract (CE) is the heterogeneous mixture isolated from soluble components of cell lysates. It contains proteins, nucleic acids, lipids, carbohydrates, and organelles from cells [43]. The CE is a cell lysate that can be obtained from all types of cells by breaking down their membranes. For many years, cell lysis has been used as a step for cell fractionation, organelle isolation, and protein extraction and purification. With the recent isolation and identification of various proteins, lipids, and genetic materials in CE, their crucial roles in regenerative medicine and in disease treatments are being reported. Currently, CE transplantation has been demonstrated as a cell-free treatment for various diseases, including wound-healing [46], myocardial infarction and ischemic stroke [47,48], acute myeloid leukemia [49], acute colitis [50], osteoradionecrosis [51], Alzheimer’s disease [52], nerve injury [53], obesity [54], liver diseases [55], Sjogren’s syndrome [56], and irradiation-induced SG injury [42] (Table 1). CE harvested from different cell/tissue sources has been analyzed, including bone marrow cells [42,51], bone marrow stem cells [56], bone marrow mononuclear cells [47], adipose stem cells [34,46], spleen tissues [34], embryonic stem cells [57], salivary gland stem cells [35], white blood cells [58], and plant stem cells [59]. Most of these showed the potential to mitigate the hypofunction of IR-injured SG.

### 2.1. Bone Marrow Stem Cell Extract and Bone Marrow Cell Extract Therapies

Mesenchymal stem cell (MSC) is first isolated from bone marrow, and bone marrow stem cell (BMSCs) has become one of the most well-studied stem cells to researchers. The extract from BMSC (BMSCE) was administered to various diseases. For example, Khubutiya et al. described BMSCE transplantation as a potential treatment for acetaminophen-induced liver failure as it reduced the area of necroses and increased the number of mitotically active cells in the liver [55]. BMSCE treatment also could preserve the exocrine function of salivary and lacrimal glands by promoting cell proliferation and extracellular matrix formation, preventing fibrosis, and regulating immunomodulation [56].

The use of whole bone marrow cell extract (BMCE) has grown in popularity due to its convenience and clinical feasibility. BMCE offers a convenient on-shelf source without the need for lengthy cell culturing. Yeghiazarian et al. isolated the BMCE and first compared its efficacy with intact bone marrow cells in a myocardial infarction animal model [47,64]. Results showed that BMCE was as effective as alive cells in reducing infarct size and cell apoptosis, enhancing vascularity, and improving cardiac function. In agreement with Yeghiazarin’s study, our previous study demonstrated that BMCE was as effective as whole bone marrow cells in repairing SG hypofunction [42]. This evidence suggested that paracrine action is the principal mechanism of cell-based therapy, and BMCE might be an alternate treatment to intact lived cell treatment. In the following years, the efficacy of the BMCE was investigated by other research groups. Michel and colleagues demonstrated that BMCE significantly enhanced new bone formation in the irradiated bone of rats [51]. A recent study revealed pain amelioration and anti-inflammatory effect of BMCE treatment in a peripheral nerve-injured mouse model [68].

Additionally, Misouno et al. found that BMCE significantly reduced focus scores in the treated NOD mice (Sjogren’s syndrome mouse model) by inhibiting lymphocytic infiltration in SGs [65]. Furthermore, this study explored the potential target proteins of BMCE by 2D liquid chromatography–mass spectrometry, including the downregulation of inflammation-related proteins (kallikrein 1-related peptidase and Calreticulin), Sjogren’s syndrome biomarker (Sjogren’s syndrome antigen B), apoptosis-related proteins (Caspase-8, CASP8-associated protein 2, and caspase recruitment domain protein 12), and the up-regulation of stem cell and development proteins (Nestin and Vimentin) and salivary gland biology markers (a-amylase, aquaporin 1 (AQP1), AQP5, parotid secretory protein (PSP). In addition to Sjogren’s syndrome, mouse and human BMCE transplantations were beneficial for IR-damaged SGs [67]. Our recent study identified the human BMCE with three cell subpopulations (mononuclear cell, granulocytes, and red blood cells) from whole bone marrow and their CEs (BMCE, MCE, GCE, and RBCE), respectively. Results showed that BMCE and MCE provided therapeutic efficacy by improving the secretory function of IR-injured SG. Both of these cell extracts did not induce an obvious immune response; GCE was of more limited efficacy but induced an acute inflammatory response. In contrast, RBCE did not restore the salivary flow rate during the observation [67]. In summary, BMSCE and BMCE, as well as specific cell extracts (MCE and GCE) derived from whole bone marrow sub-fractions, provided a promising treatment effect in SG diseases.

### 2.2. Embryonic Stem Cell, Adipose Stem Cell, and White Blood Cell Extract Therapies

In addition to bone marrow-derived CE, embryonic stem cells (ESC), white blood cells (WBC), and adipose stem cell (ADSC) are alternate cell sources for CE preparation have been investigated by researchers for disease treatments. The primary function of embryonic stem cell extract (ESCE) is the induction of differentiated cells. For example, ESCE induced to exhibit comparable properties of the ESCs [57]. Additionally, ESCE therapy on wound healing provided valuable knowledge of ESCE to promote epithelial and granulosa cells to express pluripotency markers and to undergo de-differentiation [69,70]. These differentiated cells showed the multi-potential of differentiation after incubating with ESCE. However, both studies demonstrated that ESCs-related characteristic changes were only of short duration and could not be maintained for a longer duration. The death of progenitor and functional cells is the primary mechanism during the late stage of irradiated SGs and results in a substantial loss of SG secretory function. The cell de-differentiation capability of ESCE can hypothetically mitigate SG hypofunction by promoting local SG cells to de-differentiate into progenitor cells. Nevertheless, more research is needed to test this hypothesis.

Crocodile white blood cell (cWBC) extract has been used in treating ultraviolet radiation’s effect on the skin [73]. One study revealed that cWBC extract significantly promoted cell proliferation and prevented ultraviolet-induced morphological change and skin pigmentation. Interestingly, the crocodile white blood CE induced apoptotic cell death to several cancer cell lines (including Hela, LU-1, LNCaP, PC-3, MCF-7, and CaCo-2 cells) [58,74], but no cytotoxicity towards non-cancerous Vero and HaCaT cells [58]. Although the mechanism remains unknown, this phenomenon may help patients suffering from IR-induced SG hypofunction and reducing their risk of worsening their head and neck cancers. These findings indicated that WBC extract might be a potential source for treating IR-injured SG. However, more studies are needed to investigate the immunogenicity and the optimized dosage of WBC extract before administrating it in the clinics. So far, either ESCE or WBC extracts have yet to be tested in SG diseases.

ADSCs are considered one of the most promising adult stem cells for clinical application because they can be isolated from a plethora of adipose tissues. Adipose stem cell extract (ADSCE) is widely used as a potential treatment in various diseases, such as wound healing [46], nerve injury and Alzheimer’s disease [52], obesity [54], acute inflammation [50], ischemic stroke [48] and IR-injured SGs [34]. One study reported that ADSCE reduced fibrosis and preserved the smooth muscle content in a cavernous nerve injury model. Another study administered ADSCE to Alzheimer’s disease mice and reported the antioxidant and anti-apoptosis effects of ADSCE treatment [52]. ADSCE also showed an anti-inflammatory effect on macrophage cells and suppressed LPS/IFN, induced NO, COX-2, and PGE2 production via downregulation of iNOS and COX-2 protein expression [62]. Several other studies further confirmed the anti-inflammatory effect of ADSCE [48,50]. Additionally, ADSCE was used in treating SG hypofunction. Our previous study investigated the effect of mouse ADSCE on the IR-damaged SG model. Results showed that ADSCE significantly restored the secretory function of the damaged salivary glands and protected SG functional cells, blood vessels, and parasympathetic nerves [34]. Therefore, ADSCE is expected to be used in tissue repair and regenerative medicine for SG hypofunction.

### 2.3. Other Cell Extract Candidates

Extracts derived from tissue-specific stem cells also provided a pronounced effect on the target organ treatment, such as SG stem cells. Human minor SGs can be obtained with minimal invasiveness and provide sufficient labial stem cells (LSC) for CE preparation [35]. Furthermore, SG stem cells can differentiate into various cells, especially epithelial cells [78]. Considering LSC’s extraordinary epithelial cell differentiation potential, the LSC extract (LSCE) was prepared and transplanted to rescue the hypofunction of IR-injured SGs [35]. As expected, our result showed a significant increase in the salivary flow rate of damaged SG post-IR.

In addition to the common CEs described above, animal or plant tissues as well as plant stem cells could be sources for CEs. For example, the function of adipose tissue extract was investigated in vitro with human keratinocytes, fibroblasts, and adipose stem cells. Results showed that the adipose tissue extract promoted keratinocyte proliferation and stimulated fibroblasts and adipose stem cells migration [75]. Additionally, spleen tissue extract was tested in an IR-injured SGs model, and results suggested that spleen cell extracts could mitigate SG hypofunction [34]. Plant stem cell extracts were widely used in skin anti-aging and hair loss [59,79,80,81]. Despite few studies investigating SG diseases, plant stem cell extract could be considered for SG treatment due to its diverse properties. For example, plant stem cells could promote cell regeneration and viability against senescence and apoptosis of human stem cells and delay aging [59]. Furthermore, the geranium sibiricum extract reduced the number of mast cells in the mouse skin tissue [80], and birch stem cell extract showed a suppression effect when treated with esophageal squamous carcinoma cells in vitro [82], which is a comparable inhibitory effect on cancer cells as the one observed with the crocodile white blood CE [58].

In summary, several cell extracts have demonstrated their effectiveness in treating SG hypofunction, including ADSCE, BMSCE, BMCE, MCE, LSCE, and spleen cell extracts. GCE was effective but also induced an acute inflammatory response. The optimal cell source is still unknown, and further experiments are needed to address this issue. Furthermore, other types of CEs, such as ESCE, plant cell extract, and white blood cell extract, have not yet been tested for SG diseases. They may be effective treatments for IR-injured SGs due to their capacities to repair and regenerate tissues. For example, the de-differentiated function of ESCE might benefit injured SG stem cells to renew and differentiate into functional cells (e.g., acini cells), while plant and cWBC extracts could inhibit cancer cell growth and promote epithelial cell proliferation. These promising findings open new venues for a variety of treatments for IR-damage SGs.

## 3. Optimization of Cell Extract Treatments

### 3.1. Dosage, Frequency, and Timing

Optimizing the dosage, frequency, and timing of CE administration is essential. An increase in dose might lead to a better efficacy or to a toxic effect of the therapy being tested. Na et al. demonstrated that ADSCE accelerated wound healing by promoting dermal fibroblast proliferation, migration, and extracellular matrix production [46]. While Lim et al. compared the therapeutic effect of ADSC and ADSCE and demonstrated that ADSCE showed no effect on wound healing in a mouse model [60]. One possible reason was that the dosage in Lim’s study was much lower than that in the former (Table 2). This indicates that the therapeutic effect of ADSCE may be dose-dependent, and that the dosage of each CE should be titrated in animal models prior to clinical studies. Notably, the actual dose might be affected by other factors and varied in different situations. For example, the more effective the delivery method, the larger the dose can be available to the targeted tissues and cells. Therefore, all the factors (such as delivery approach, injection frequency, and the combined effect with other drugs) associated with dosage should be considered.

Timing is another vital factor that should be considered in CE treatment planning. A study reported the effect of anti-aging and weight gain reduction following MSC extract (MSCE) treatment [54]. However, another study by Hsu et al. demonstrated that MSCE treatment accelerated osteopenia and lymphopenia and consequently produced a cachexia-like effect to decrease longevity in aging rats [63]. The authors also suggested that the cell lysate treatment did not affect the longevity and vitality of middle-aged rats. This finding suggests the importance of selecting the appropriate age of patients for a specific CE treatment. On the other hand, the timing of starting treatment also influences the therapeutic effect. For example, starting BMCE injections between 1–3 weeks post-IR mitigated the hypofunction of IR- induced injury to SG, while starting BMCE treatment after seven weeks post-IR did not restore the secretory function [83]. One reason might be that the active ingredients in BMCE have limited abilities to rescue the functional cells in SGs at the chronic stage (>30 days) once they have been damaged irreversibly following the IR injury. In contrast, CE treatment might mitigate the damage of SGs in the first 1–3 weeks post-IR. Regarding the mechanism of IR damage, this evidence indicates that CE may prevent the M3-muscarinic receptor and the water channels in the plasma membrane from the IR damage and further protect the functional cells, such as the acinar cell, thus restoring the function of damaged SG. This hypothesis needs to be verified in the future. More investigations should focus on selecting treatment timing for the individuals and the mechanism behind it.

The frequency of BMCE treatment has been investigated in our previous study [83]. We found that the therapeutic effect of BMCE lasted eight weeks post-treatment and could remain for a longer time by increasing the frequency of injections per week. The study demonstrated that multiple injections helped to keep the concentration of the effective constituents over a minimally adequate level and maintained a favorable microenvironment for cells in SG. Another recent study is consistent with our findings. Nishikawa et al. administered ADSCE to mice with acute colitis. They demonstrated that the anti-inflammatory and anti-apoptosis effects were only detected in the group with the injection of ADSCE and lasted for three successive days, but not in the group with a single administration [50]. This evidence indicated that the benefit of CE treatment depends partly on the administration frequency.

### 3.2. Delivery Methods

Three delivery routes have been used to deliver CEs in preclinical studies, including the local [42], intraperitoneal (IP) [52,84], and intravenous injection (IV) [56]. There are advantages and disadvantages of each method (Table 3). Our group compared intra-glandular (IG) versus intravenous injection and found that both methods significantly increased the secretory function of submandibular glands post-IR [42]. We also noted that there was no difference in treatment effect between one-time IG and four-time IV injections. Consistent with our results, a localized injection (intracavernous injection) of adipose stem cell extract reduced fibrosis and recovered nerve fibers in a rat cavernous nerve injury model [61]. These results indicated that both local and intravenous CE injections were appropriate methods in animal studies. In contrast, a study of bone regeneration showed no significant bone formation post-in-situ injection of MSC-extract/lysate in a beagle dog model [71]. Another research group compared different delivery approaches and revealed that intravenous injections of BMCE enhanced bone regeneration post-irradiation, while the intraosseous injection group did not [51]. One possible reason is that the soluble factors in CE lost their active effects in the osseous microenvironment affected by irradiation or trauma (e.g., hypo-vascularized environment). Moreover, these two studies are all based on a bone regeneration model. It indicated that the localized delivery method might not be appropriate for the bone-damaged model, but it is still a potential route for other animal models.

Clinically, local delivery methods, including IG and cannulation of retrograde ductal cannula instillation, are feasible methods recommended for SG treatment [55,85]. The retrograde infusion method is being used in animal models and in clinical trials to restore the function of IR-induced SG hypofunction [84,86] as it is a non-invasiveness technique. However, this method is challenging to perform in small animal models, such as the mouse model, while the IG can be the alternate approach in a preclinical study. Recently, IG guided by ultrasound was performed in several clinical trials with adipose stem cells and effective-mononuclear cell transplantations for patients with IR-Induced xerostomia [40,87].

### 3.3. Cell Extract Preparation

Over the past decades, a standardized isolation method for CEs has yet to be available; different techniques have been developed considering the cell’s biochemical and physicochemical features to lyse the cells and prepare the CE. However, until now, only some studies have systematically reviewed all the methods of CE isolation. Herein, by comprehensively summarizing and analyzing the progress of CE preparation techniques, we provide an overview of CE isolation strategies and their related applications. Four basic cell lysis methods were commonly used for CE preparation, including ultrasonic homogenization, temperature treatments, osmotic and chemical lysis methods. Researchers usually combine one or two methods to confirm the lysis effect. The combination of the lysis methods is summarized in Table 2. In brief, there are five protocols reported in current CE studies: (1) ultrasonication method, (2) lysis buffer (chemical lysis) + ultrasonication method, (3) distilled water (osmotic) + ultrasonication method, (4) freeze and thaw cycle method (temperature treatment) and (5) distilled water + freeze–thaw cycles methods (Table 4). Although these combinations are based on the classical disruption techniques, the cell extract’s final constituents and activity vary and may become a challenge for future clinical trials.

Even if research groups used the same combination methods for CE preparation, the protocols varied between research groups, which may affect the cell extracts’ components. For the temperature method, as an example, regarding the duration of the freeze and thaw cycles (from two to four cycles), a consensus has yet to be reached [35,50,71]. Apart from the apparent differences in the primary extraction method, other details of the process can also interfere with the ingredients, properties, and effects of the CE. Centrifugation is necessary for the final purification of CE to remove the insoluble materials. This separates membranous particles by size and buoyant density [89]. Centrifugation speed and time range between research groups from 12,000× *g* to 17,000× *g* and 15 min to 30 min [34,46,54,90]. Most studies’ rotation speed and duration vary between 1000× *g* to 100,000× *g* and from 10 to 60 min [49,62,63]. These parameters may isolate different “soluble materials/components” in the CE. For example, 1000× *g* for 10 min removes nuclei and cell debris from CE and 15,000× *g* for 10 min can remove mitochondria, lysosomes, peroxisomes, and rough endoplasmic reticulum (ER), while, in comparison, 100,000× *g* for 60 min can pellet the microsome (smooth/rough ER, exocytic and transport vesicles, plasma membrane, Golgi, and endosomes) [89]. It suggested that a consensus on the isolation method, the definition of the particle size, and the significant ingredients of soluble materials in CE needs to be reached. Other details, such as the storage method (lyophilization), filtration, and the recipes of the lysis buffer, may also influence the constituents of the CE.

The most common method of preparing CEs to treat SG hypofunction, such as Sjogren’s syndrome and IR-damaged SG, has been the freeze–thaw cycle technique [42,43,66]. As to why investigators selected this (temperature) strategy, we hypothesize that the freeze–thaw method is safer and more stable than other methods for preparing the CE of mammalian cells and for administrating it in preclinical or clinical studies. For instance, adding additional chemical components is unnecessary compared to the chemical cell lysis method, which theoretically results in a purer CE product. An added advantage of the freezing–thawing method is that an isotonic solution can be used during the procedure. Therefore, compared to the osmotic method, the protein lysed from the cells would stay in the buffer (such as PBS or normal saline) and become more stable when dissolved in distilled water. Furthermore, a study has found a positive effect of a small amount of sodium chloride: it can help dehydrate the protein better via water uptake and facilitate proteins refolding into their stable conformation during lyophilization [89,91,92], thereby increasing the stability of proteins.

In conclusion, it is necessary to compare the constituent and the effect of the CEs prepared by different methods and optimize these combination techniques to increase the efficacy and stability for both the disease treatment and the isolation methods of CEs. It is also worth noting that not all biological material isolation requires the same technique. This is because a single technique that fits a variety of tissue and cell sources is not practical. For example, mechanical homogenization and ethanolic extraction would be an additional procedure for preparing CE derived directly from tissue or plant (but not mammal) cells [34,76,80,81]. One reason is that a rigid cell wall surrounds the plasma membrane of plant cells, while cells in animal tissue are surrounded by an extracellular matrix, thereby increasing the difficulty of cell lysis. Putting this together, efforts should be made to exploit several standardized protocols for different CEs. Additionally, investigators should carefully choose the most appropriate method to yield the optimum effect for their studies.

## 4. Constituents in Cell Extracts and Their Mechanisms

CE is a group of heterogeneous mixtures containing inter-cellular materials, such as DNA, RNA, protein, lipids, and organelles from cells [43,88]. Recently, CE has been tested for regenerative medicine in various of disease models in vivo and in vitro. However, the active components in CE have yet to be fully identified. To this end, the first step is to determine which categories of molecules in a CE, for example, nucleic acids or proteins, contain the bioactive factors. The second step is to compare the constituents in CEs from different cell sources, such as adipose tissue or bone marrow. The third step is to profile the ingredient in CEs. The last step is to explore specific candidate factors in the CE responsible for the therapeutic effect and unveil their mechanisms of action.

In a previous study, we answered the first step by uncovering that the effective bioactive factors in the CE were proteins [43]. We deactivated proteins in the CE by using the proteinase K combined and then heating at 95 °C to inactive proteinase K before injecting this deactivated CE into mice. Then, either normal saline, CE, or deactivated CE were administered in the IR-injury SG mouse model. Results showed that the deactivated protein CE injected was no better than the injection of saline, while the infusion of native CE restored the secretory function of SG. This finding indicated that the native proteins (but not the DNA, RNA, lipids, carbohydrates, or other small organelles) were the effective constituents in CE.

Analyzing the different constituents (proteins) in diverse CEs was tested next. Apart from the difference induced by various isolation methods, the type of cell source is another major factor influencing the components of CEs. A study compared the therapeutic effect and the constituents of the spleen, adipose, and bone marrow CE [88]. All three CEs restored the hypofunction of IR-injured SGs and protected the functional cells, blood vessels, and parasympathetic nerves during the 8-week observation. Preliminarily, to analyze the protein components in CEs derived from different cell sources, a protein membrane array assay was used to profile angiogenesis-related factors in these three CEs. Results showed that the constituents and concentrations of certain growth factors differed between CEs. For example, a significantly lower SDF-1 was detected in the spleen CE in contrast with bone marrow and adipose CE. The adipose stem cell extract presented a higher number of angiogenesis-related factors than other CEs. One interesting finding is that the ADSCE showed less efficacy than the other two CEs, although it contained the most identified growth factors in this study. Aside from this difference, several overlapping proteins were identified in these CEs, such as MMP-9, CD26, and OPN. Another recent study confirmed these findings and showed that the concentrations of identified proteins in the mononuclear, granulocyte, and red blood cell extract from human bone marrow were different [67]. However, unlike ADSCE, mononuclear cell extract contained more growth factors and provided the best therapeutic efficacy in treating the IR-injured SG. This study further compared the bone marrow cell extract from different species (mouse and human) and demonstrated that more angiogenic factors were detected in human BMCE. Altogether, CE derived from different cell/tissue sources contained several overlapping proteins and certain different constituents. This difference may influence the therapeutic efficacy of disease treatment.

Admittedly, only some studies systematically profile the stem cell extracts with comprehensive proteomic analysis, but several CEs had been semi-quantified by protein membrane assays. One of the studies presented 171 cytokines identified from ADSCE [63], including angiogenic factors (FGFs, VEGFs, ANGs), tissue remodeling proteases (MMPs) and its inhibitors (TIMPs), stem cell homing chemokines (SDF-1), anti- and pro-inflammatory cytokines (IL-1β, 6, 8, 11, 17, IL-1ra, and TGF-β1), tissue repair/regeneration-related factors (BMPs, IGFs, PDGFs, and HGF), and many other cytokines. Three categories of proteins in mouse bone marrow cell extract were preliminarily screened, including angiogenesis, cytokines, and chemokines [35,43]. Several angiogenic factors were identified, such as CD26, FGF, HGF, MMPs, PF4, and SDF-1, while few cytokines (IL-1ra and IL-16) and chemokines were detected in mouse BMCE. There were 22 angiogenesis-related growth factors detected in human BMCE [67] and 26 were found in the human labial gland stem cell extract [35]. One study profiled the human adipose tissue cell extract (adipose liquid extract) through proteomics (Figure 1) [76]. A total of 1742 proteins were identified in the adipose tissue cell extract, most of them were from the cytoplasm (62.2%), followed by the nucleus (16.3%), extracellular space (9.3%), and plasma membrane (8.7%). These molecules were mainly involved in the cellular process, biological regulation, and metabolic process. These results demonstrated that the CE contains crucial growth factors and cytokines related to numerous physiological and pathological pathways in our body, and this is perhaps the reason for the broad systemic effect of the CE in treating a variety of diseases.

Different cell extracts might contain several overlapping proteins, such as FGF-1, -2, MMP-8, -9, VEGF, TIMP-1, CD26, PAI-1, and SDF-1 [35,43,63,67]. Many of these are multifunctional and play a role in treatment. FGF-2 is a highly expressed factor in the labial stem cell extract [35], bone marrow cell extract [67], and adipose stem cell extract [34,63]. It is a well-known mitogen that plays an essential role in angiogenesis and wound healing. FGF2 promotes fibroblasts and the epithelial and endothelial cell proliferation and induces the regeneration of tissues and blood vessels [93,94]. It also encourages the growth of acinar cells [95], myoepithelial cells, ductal cells [93], and the development and regeneration of salivary glands [93,96]. Studies reported that FGF2 protected IR-injured SGs by inhibiting radiation-induced apoptosis in vivo and in vitro [95,97]. Stromal cell-derived factor-1 (SDF-1) is another critical factor in CE that plays a role in the disease treatment [98], with a potent capacity to repair damaged tissues by regulating immune response, inflammation, cell migration, vascularization, and neurogenesis [98,99,100]. Interestingly, besides the angiogenesis factors, the anti-angiogenesis factors (such as TIMPs and PAI-1) were detected in CEs [35,63,67]. These factors generally inhibit cell proliferation, migration, and angiogenesis [101,102,103,104,105,106]. However, they showed diverse benefits in treating IR-induced SG hypofunction. For example, most of these anti-angiogenic factors are natural inhibitors of the tumorigenesis [102,103,104], which is the main advantage for head and neck cancer patients. Fang et al. reported that CE from bone marrow contained pro- and anti-angiogenesis factors that did not promote tumor cell proliferation [34,43]. The modulated interactions of these anti-angiogenic growth factors might be part of the reasons explaining this underexplored phenomenon.

Besides the effect of a single molecule, the interrelation between co-existed factors in a CE is a concern during treatment. VEGF, an angiogenic factor with neurotrophic and neuroprotective effects [107], was identified in multiple CEs as FGF-2. Studies reported that other angiogenic factors were required to complement VEGF to promote vessel maturation because neo-vessels were unstable with the sole use of VEGF [108,109]. This implies that the synergistic effect of diverse proteins played a role during treatment and the therapeutic effect was attributed to the interactions of a variety of growth factors in CE rather than one or two vital factors. Apart from this synergistic effect, antagonist interactions of growth factors were also found in CEs. PAI-1 acts as an anti-angiogenesis factor and an inhibitor of urokinase (uPA), but both of them co-exist in labial stem cell extract [35]. Studies reported that a fine balance of cell migration, wound-healing, embryogenesis, and angiogenesis happened between the PAI-1 and uPA regulation [110]. Similarly, our previous study revealed that the therapeutic effect of labial stem cell extract was attributable to the interactions of pro- and anti-angiogenic factors in the CE [35]. A similar relationship can be found between TIMPs and MMPs in CEs [67]. Altogether, these results suggest that the therapeutic effects of different CEs are modulated by multiple factors as well as their interactions (synergism and antagonism). These factors are an essential element of the SG repair and regeneration process. More studies need to be carried out in order to decipher these complex interactions.

Indeed, instead of targeting a key specific molecule, CE treatment is a category of treatment given for a broad systemic change with a well-orchestrated cascade. Sam Zhou et al. reported that bone marrow CE partially restored serum proteomic homeostasis and re-established systemic balance to attenuate mechanical hypersensitivity in a nerve injury mouse model [68]. Interestingly, no strong regulation was detected in the serum of CE-treated mice. These results suggested that CE treatment tends to regulate systemic homeostasis but not that of a single molecule. In addition, the serum from CE-treated nerve-injured mice no longer induced hypersensitivity in naïve mice. This finding verified that CE treatment, as a multifaced systemic approach, alleviated pain by causing a broad modification in the serum of the treated mice. Overall, we propose that the mechanism of CE treatment is complicated and comprehensive and that it tends to establish a new systemic balance by functional molecules and their interactions in CE to improve the pathologic microenvironment and consequently alleviate diseases, such as the IR-injured SG.

It is worth noting that the group of active factors in CEs remains unknown. Further studies are required to verify and purify the effective molecules in CEs (or separate the inactive substances from the CE) and unveil the different mechanism pathways behind them. Our recent study makes an effort to separate the subpopulations of bone marrow cells and compares the effect of three CE fractions (the cell extract from the mononuclear cell, granulocyte, and red blood cell) [67]. Results showed that the mononuclear cell extract provided the best therapeutic efficacy, while the red blood cell extract did not significantly mitigate salivary hypofunction. This implied that the effect of the bone marrow CE treatment could be improved by removing the red blood cell from bone marrow. On the other hand, this finding suggests that the “purification” of the CE might help to obtain an optimized CE with the best therapeutic effect. Lastly, individual differences, including age, gender, and physical condition, might influence the cell extracts’ constituents and treatment effects. Therefore, it is important to clarify the difference by comparing the CEs from many of donors. Individual differences should be considered by the researchers when preparing the CEs from human cells or tissues and utilizing them in the clinic.

## 5. Advantages and Limitations

Bioregenerative therapies have drawn much attention in recent years. Cell-free therapy and cell-based therapy are considered the prospective regenerative treatments applied in clinic. Cell-free therapy, including cell extract (CE), conditioned medium (CM), extracellular vesicle (EV)/exosome, and platelet-rich plasma (PRP), are reported to be multifunctional in disease treatment. This section will discuss these therapies for IR-injured SGs and compare them with cell extract treatment. The advantages and limitations of cell extract are also discussed.

### 5.1. Cell Extract and Cell-Based Therapies

Numerous cell-based therapies have been reported as potential regenerative therapies to relieve xerostomia caused by radiotherapy [33,36,37,41,49,111,112]. Woodward et al. demonstrated that bone marrow stem cells secreted growth factors, mitigated the immune response, alleviated inflammation, and promoted the remaining local stem cells to differentiate and proliferate in SGs [113]. Other adult tissue-specific stem cells, such as adipose and SG stem cells, also showed potential in treating salivary hypofunction. Three studies have revealed that adipose-derived stem cells alleviated the hypofunction of the salivary gland post-irradiation in mice [112,114] and rat models [115]. Recently, adipose stem cells have been tested in phase I–II clinical trials [40]. The stem/progenitor cells in the salivary glands are another candidate cell source for treating irradiation-induced. Dr. Coppes’ group isolated and cultured murine SG cells into the spheres and reported that the cells were positive to stem cell markers, such as CD117, CD24, CD29, CD49f, CD44, CD90, CD34, Sca-1, Mushashi-1, and c-Kit [116]. Then, the human SGs-derived spheres were successfully formed with the cells positive to c-Kit [117,118,119]. These c-Kit positive stem cells could self-renew for more than 48 weeks in vitro and in vivo [116,117,120,121]. Furthermore, the c-Kit^+^ cells rescued hyposalivation and maintained the homeostasis of SGs post-irradiation [116,121]. More recently, the studies concerning SG c-Kit^+^ cells were tested in phase I–II clinical trials [40,117,122]. In addition to the c-Kit^+^ cells, there is another tissue-specific stromal/ mesenchymal stem cell isolated from human major SGs [123,124] capable of multiple differentiation, including osteogenesis, adipogenesis, and chondrogenesis and could generate epithelial cell types (epithelial and hepatic cells). Furthermore, these stem cells showed the capability to treat SG hypofunction induced by irradiation. Apart from the stem cells, several other cell sources have been studied in SG regeneration, such as bone marrow cells [38], peripheral blood mononuclear cells [39], and dental pulp cells [36]. Additionally, there are currently other potential cell sources, such as Wharton jelly stem cells [125], menstrual blood-derived cells [126,127], and bone marrow-derived mononuclear cells [128,129,130], which exhibit their potential in treating many other diseases. However, limited evidence shows their application in SG repair and regeneration.

The paracrine cytoprotective effects of cell-based therapy were reported in various diseases. One research group cocultured hypoxic human adipose stem cells with IR-injured SG epithelial spheroids in vitro and showed that hypoxic conditions increased the therapeutic effect by promoting stem cells to release more growth factors (e.g., FGF10) and activating FGFR-PI3K signaling [37]. Our previous study showed that cell extract from bone marrow was as effective as bone marrow cells in restoring the secretory function of SGs damaged by irradiation [131]. These results demonstrate a paracrine mechanism of action from stem cell-based therapies and thus also support the achievement of the therapeutic effects of cell-free treatments.

Stem cell transplantation is an efficacious approach to restore SGs functionally. However, there are several limitations in the clinical use of cell-based therapies which need to be taken into account, such as limited cell lifespan (e.g., passage 3–5) [132], the potential risks of tumorigenesis and immunoreaction [43], and the low efficiency of the engraftment of the transplanted cells [39,133,134,135,136]. Thus, cell-free therapies (such as cell extracts) are an alternative approach to overcoming these limitations associated with cell-based treatments.

Cell extract treatment, used as a type of cell-free therapy, can potentially treat SG hypofunction. Accumulating evidence indicates certain advantages of cell extract injection compared to cell-based therapies. First, the injection of cell extract elicits a smaller immunoreaction in the recipient animals or patients [137]. Studies reported that human bone marrow and salivary gland cell extract treatments improved the function of damaged SGs without a severe immune reaction in an immune-competent mouse model [35,67]. These results agree with our previous study, which demonstrated that the cell extract contained fewer histocompatibility antigens than the intact cells and resulted in a weaker immune response due to the cell-free agents [43]. In addition, we have reported that cell extract treatment is not patient-specific [42]. An added advantage of cell extract is that variables such as male or female donors or autologous or allogeneic would not affect the therapeutic effect of the treatment [53,67]. Hence, a cell-free therapy (cell extract treatment) provides the possibility of both autologous and allogeneic transplantations. Second, cell extract treatment theoretically has fewer risks associated with the possibility of tumor formation. Cell-based treatment can potentially stimulate cancer cell growth or cell differentiation into cancer cells, while cell-free treatment is considered theoretically safer in this sense [42,43]. In addition to safety, the administration of cell extract is more feasible for clinical application. For example, cell extract can be cryopreserved in a −80 °C freezer for 12 months without protein degradation [34]. This demonstrates that cell extract could be simply stored for up to one year and be ready to use at any time. Moreover, our previous study developed a practical lyophilization technique for long-term cell extract storage without the loss of product potency; it further prolongs storage time and reduces cost, thus extending the application of cell extract treatment in the clinic [66]. In contrast, it is difficult to treat with live cells at any given time due to requirements such as cell cryopreservation, cell-thawing, recultivation, and obtaining sufficient live cells at the appropriate passages. As a result, it is time-consuming to prepare the cells before each treatment, while the cell extract is easy to manage and use. Altogether, we propose that cell extract therapy is a safer, more practical, and more economical approach when compared to cell-based therapies.

### 5.2. Cell Extract and Other Cell-Free Therapies

#### 5.2.1. Conditioned Medium Therapies

Conditioned medium (CM), also known as the secretome, is a complex mixture secreted by live cells into the extracellular space and contains soluble proteins (growth factors, cytokines, chemokines, enzymes, signaling and signal transduction proteins, and cell adhesion molecules), nucleic acids (DNA, RNA, and microRNAs), as well as lipid molecules and extracellular vesicles (EV) (apoptotic bodies, micro-vesicles, and exosomes) (Figure 2) [138,139,140,141]. The secretome in CM can affect several functions, including vascularization, tissue differentiation, metabolism, defense response, hematopoiesis, and skeletal development [142]. The concentrations and composition of CM varies between cell types. For example, a high level of IL-6, TGF-β1, and IGF-1 was detected in adipose stem cell CM [44,143], while almost no IGF-1 was identified in the CM derived from periodontal ligament stem cells [144]. In addition, the constituents in CM varied if the cell culture condition changed, such as the three-dimensional (3D)/spheroids culture with or without diverse scaffolds [138,145], the hypoxic condition [37], or added supplement stimuli in different culture medium [146]. Shin et al. reported that the hypoxia-activated MSC prevented IR-induced SG hypofunction by enhancing the paracrine effect of FGF10 [37]. A recent study compared the secretome from two- and three-dimensional (2D, 3D) MSC culture and reported that both secretomes restored the histological structure of acutely injured lungs and decreased fibrin deposition, but the 3D group exhibited a more pronounced trend in lung recovery [145]. Then, they further investigated the molecular in CM by proteomic with liquid chromatography–tandem mass spectrometry (LS/MS/MS). A total of 281 and 286 proteins were identified in 2D and 3D CM groups, and 47 and 52 proteins were exclusively included in 2D and 3D CM, respectively. In addition to the cultivation condition, the difference in the isolation method of CM results in different components. For example, the culture duration of CM collection, varying from 16 h to 120 h, might induce different secretome products [139]. One probable reason is that the cell state might change due to a longer period of cultivation with a serum-free medium. This condition might inhibit cells to release serum-acquired molecules. These results suggested that the treatment effect could be improved by changing the cell culture condition or improving the isolation procedure of CM. Therefore, optimizing the culture conditions is crucial for obtaining the best secretome.

CM showed promising results in the treatment of numerous diseases, including wound-healing [147], Huntington’s disease [148], spinal cord injury [149], colitis [150], liver diseases [151], periodontal disease [144], and IR-injured SGs [44]. An et al. reported that human adipose MSC secretome contained high levels of VEGF, IGF-1, and GM-CSF that strongly induced cell proliferation and salivary proteins, thereby remodeling the damaged SGs [44]. Hypoxic conditions and co-culturing with platelet-rich fibrin promoted the repair of damaged SGs [44,152]. Therefore, conditioned medium treatment is a potential therapy for salivary hypofunction. Further experiments are required to identify and quantify the dynamic expression profile of each conditioned medium, its functional factors, and related signaling pathways for disease treatments.

#### 5.2.2. Other Potential Cell-Free Therapies

In addition to the CE and CM, several other potential cell-free therapies could be used in treating IR-damaged SG. Herein, platelet-rich plasma, extracellular vesicles, and exosomes will be introduced in this section with potential properties for treating the IR-injured SG.

Platelet-rich plasma (PRP) derived from a whole blood sample can be obtained simply through a specific centrifugation process to separate the fatty cells effectively (such as the white blood cells, red blood cells, and stem cells) and PRP (lighter platelets and plasma) into distinct layers.

Numerous researchers have shown excellent outcomes of PRP administration as a cell-free therapy for various diseases, such as osteoarthritis [153,154], acute muscle injuries [155], periodontal disease [156,157], dry eye, and corneal ulceration [158]. In addition to the preclinical studies, PRP treatment was performed in many clinical trials, relieving patient symptoms caused by osteoarthritis and demonstrating a strongly effective outcome [153]. Furthermore, PRP treatment’s maximum effect lasted 180 days after a single injection [159]. A combination of PRP with other supplementals, such as stromal vascular fraction [160], MSC [161], and hyaluronic acid [162], significantly increased the efficacy of PRP treatment to alleviate the symptoms, suggesting that the combination treatments might be a better option for PRP application [153]. One recent study reported that PRP and MSC suspended in PRP successfully regenerated the cells in SGs and that the MSC combined with the PRP group provided the best results [163]. This study exhibits PRP as a potential product to treat SG diseases. One added advantage of PRP is that it can be easily obtained from patients. This property makes autologous transplantation to each patient possible due to its low invasion and wide acceptance by the patients. However, there is a study that revealed that that transplantation of PRP with MSCs failed to provide a better therapeutic effect than administering cell-based therapy only [154]. Furthermore, combining PRP with cell treatment essentially becomes a cell-based therapy in which more investigations are required to confirm the efficacy of PRP treatment only or optimize PRP treatment with other safe and practical supplements.

Extracellular vesicles (EVs) are the particles naturally secreted by most cell types, usually classified by their size, biogenesis, and functions. The three main subtypes of EVs are named exosomes (50–100 mm in diameter), microvesicles (100–1000 nm in diameter), and apoptotic bodies (1000–5000 nm in diameter) [164,165]. EVs enriched with important bioactive molecules can deliver proteins, RNA, DNA, lipids, and carbohydrates to their target cells, resulting in information exchange and host cell reprogramming [164]. Furthermore, diverse molecules in EVs are involved in maintaining tissue homeostasis and a wide variety of biological functions, such as anti-fibrosis, cell proliferation, migration, angiogenic and anti-apoptotic function regeneration, and immunoregulation [156,166,167,168]. The exosome is the most prominent component in EVs. It originates from the intracellular budding of endosomes and is released from an original cell into the extracellular space [169,170,171], involved in regulating the phenotype, function, survival, and homing of cells [164]. In addition, exosomes can transfer biological information over long distances to their target cells to elicit pleiotropic responses [172,173]. Thereby, exosomes play a vital role in intercellular communication, signal transduction, and other paracrine mechanisms in vivo [174]. A study demonstrated that both the cell-based therapy and the cell-derived exosomes showed similar miRNA profiling, indicating the crucial role of exosomes in the treatment [174,175].

Both EVs and exosomes are a prospective cell-free therapy, according to their promising potential and multiple properties, and have been involved in treating a variety of diseases, such as liver disease [170], neurological disorders [169,176,177], irradiation-induced lung injury [168,178], kidney diseases [179], cardiac diseases [180], and acute and chronic skin wounds [181]. However, only a few have been administrated in the treatment of diseases of the salivary glands. One recent study demonstrated that EVs from adult stem cells showed potential for SG tissue regeneration, particularly for angiogenesis and neurogenesis [182]. Other investigations have reported that salivary gland organoid-derived exosomes significantly simulated epithelial growth and mitosis, as well as epithelial progenitors and neuronal growth in IR-injured SGs ex vivo [183]. According to these outcomes, we propose that EVs and exosomes are excellent candidates for treating IR-damaged SG.

Admittedly, there are challenges regarding EV and exosome administration. One limitation is the low yield of EVs/exosomes due to the limited secretion capability of cells. Although research has contributed to improving the yield by developing cell culture methods [184], the difficulty of large-scale clinical applications is still extant. Another challenge is that the purification and isolation of EVs, especially the exosomes, still need to be standardized. Additionally, the active factors in EVs/exosomes have yet to be identified. Lastly, safety assessments, such as cytotoxicity and side effects, also require clarification. For example, using exosomes in the treatment of tumor progression is controversial [185]. Several studies demonstrated the suppressing impact of exosomes by inhibiting tumor cell proliferation and promoting apoptosis [186,187,188], while others reported that MSC-derived exosomes promoted tumor growth [189,190]. Altogether, further investigations are required to verify the efficacy of EV and exosome treatments to restore the secretory function of SGs.

#### 5.2.3. Comparison of Cell Extracts to Other Cell-Free Therapies

Cell-free therapeutic strategies, such as CM, PRP, EV/exosome, and CE possesses, have many similar properties and advantages when compared to (live) cell-based therapies (Table 4). For example, tumor formation and immune rejection risk is considered lower in cell-free therapy transplantation. Cell-free therapies can be stored in the freezer and are thus more practical for clinical application than cell transplantation. Nevertheless, there are differences among cell-free derivatives. First, CE is enriched with soluble materials obtained through physical isolation strategies, such as the freeze–thawing method, and does not require the addition of other chemical materials, unlike CM, thus significantly reducing the risk of biological contamination and promoting safety for future clinical applications. Second, preparing CM, EVs, or exosomes requires cell isolation and cell culture procedures, while cell extract from tissues or cells (such as bone marrow cells or white blood cells) does not require cell cultivation in vitro and can be quickly and easily obtained in the laboratory or even in the operating room [67,73,77]. Third, the efficacy of single treatment PRP (not combined with cell-based therapy) in SG diseases or other diseases is still being determined. In contrast, other cell-free therapies could be used as an effective treatment independently without the support of cells. Fourth, because of fewer preparation procedures, CE would reduce cost and time and consequently be more economically profitable for patients and clinicians. Lastly, the number of cells required for the other cell-free therapies to obtain sufficient bioactive factors is much more than that needed for CE preparation. The isolation of a higher yield of bioactive factors in CE makes this product possible for clinical regenerative medicine.

### 5.3. Limitations of Cell Extract Treatment

Despite significant advances and promising results in cell extract therapy for SG repair and regeneration post-IR, several fundamental questions and challenges associated with its clinical application need to be answered. One of the significant limitations is that the bioactive molecules in cell extract are still unknown. Therefore, a comprehensive profile of these molecules in cell extract is required. Second, the cell extract consists of numerous molecules, and there is a possibility that only some of these ingredients are effective molecules for disease treatment. Therefore, narrowing down the complexity of the cell extract by purifying the bioactive factors specific to the desired tissue targets might be an approach to reduce side effects and promote therapeutic outcomes. Third, cell extract is known to exhibit features of their specific parental cell types. Thus, it would be imperative to ensure that cell extract does not carry any functional limitation from their cell sources, such as cell status or the donor’s age. Fourth, there are various methods of cell extract preparation and different delivery routes for cell extract administration. Thus, optimizing and standardizing cell extract isolation and transplantation approaches is essential. Fifth, multiple injections might be required to maintain the desired effect of cell extract treatment [83]. However, the optimal frequency and dosage of the cell extract treatments have not yet been found. Lastly, to bring cell-free treatment a step closer to clinical reality, the safety and efficacy of cell extract treatments need to be assessed extensively and clinical trials using cell extract on IR-injured SGs need to be performed (Table 5).

## 6. Conclusions

Cell-free therapy overcomes the limitations of conventional and cell-based therapies for SG diseases and show numerous advantages, such as therapeutic and economic efficacy, less invasiveness, convenience, and safety. Cell extract treatment, as a cell-free derivative, has been shown to be promising in preclinical studies for IR-injured SGs. Cell extract with specific properties and effects can be isolated from various cells and tissues. It provides a high possibility of satisfying a variety of situations of individuals to achieve autologous and allogeneic transplantations in the clinic. Regarding the safety, feasibility, and efficacy of cell extract treatments, there are still questions and challenges that need to be solved before its clinical translation, such as the optimization of injection routes, timing, frequency, and dosage; the standardization of isolation techniques; the selection of the most effective and appropriate cell extracts; and the extension of the knowledge of the molecular profile in cell extracts. Therefore, further improvement is needed for cell extract administration. To conclude, cell extract therapy has broad clinical application prospects and could be an alternative treatment approach for SG hypofunction induced by IR.

## Figures and Tables

**Figure 1 biology-12-00305-f001:**
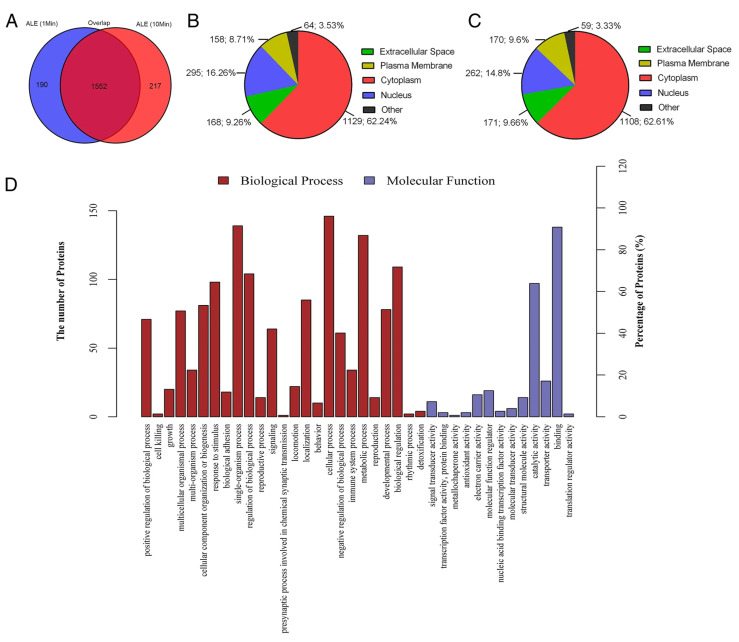
Proteomic profiling of ADSC extracts using mass spectroscopy. (**A**) Expression profile of proteins from ADSC lysate with 1-min and 10-min processing time. (**B**,**C**) Subcellular distribution of proteins in 1-min and 10-min processed samples, respectively. (**D**) Biological and molecular functions of the identified proteins. Reprinted with permission from He et al., 2017 [77]. Figure reproduced under Creative Commons Attribution 4.0 International License (http://creativecommons.org/licenses/by/4.0/ (accessed on 2 February 2023)) and Creative Commons Public Domain Dedication waiver (http://creativecommons.org/publicdomain/zero/1.0/ (accessed on 2 February 2023)).

**Figure 2 biology-12-00305-f002:**
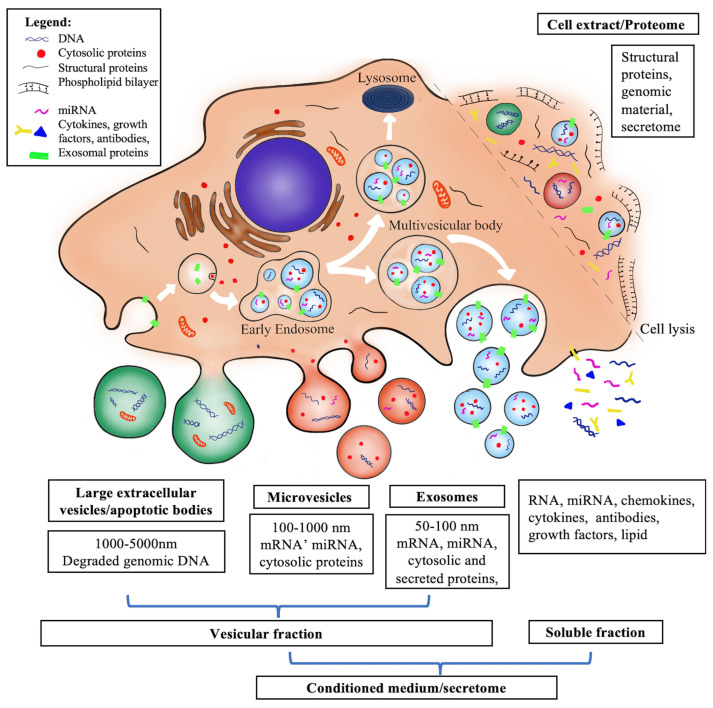
Mesenchymal stem cell-derived cell-free therapies and their components. Various particles are secreted from cells during function, bundled as vesicles or as soluble factors directly in the extracellular space. Vesicular fraction contains apoptotic bodies (1000–500 nm), microvesicles (100–1000 nm), and exosomes (50–100 nm). Apoptotic bodies are formed in response to apoptotic signals through the budding of the cell membrane and contain genomic material and mitochondria. Microvesicles are also created by the outward budding of the cell surface, while exosomes are secreted by the inward blebbing of the endosomal membrane (White arrows). Microvesicles and exosomes can contain receptors, transcription and transduction factors, enzymes, growth factors, lipids, and nucleic acids (DNA, mRNA, miRNA). The soluble fraction contains proteins, growth factors, and chemokines. Cell secretome or conditioned medium comprise of all the above. On the other hand, cell extract contains the cell secretome and the proteome (structural and cytoplasmic proteins, genomic material, and organelles) as it is obtained by whole cell lysis.

**Table 1 biology-12-00305-t001:** Cell extract (CE) treatment for various diseases.

Condition	Results/Effects	Author/Year	Ref.
Adipose Stem Cell Extracts			
Wound healing	Better wound-healing through cell therapy than CE	Lim, 2010	[60]
Wound healing	Dermal fibroblast proliferation, migration and ECM formation observed (Collagen type 1, MMP1)	Na, 2017	[46]
Nerve crush injury	Reduced fibrosis, increased smooth muscle content, and improved erectile function	Albersen et al., 2010	[61]
Nerve crush injury	Improved erectile function in both autologous and allogenic CE transplantation	Mangir et al., 2014	[53]
Inflammation	Anti-inflammatory effects seen on macrophage cells in vitro, reduction of Nitric Oxide and COX-2 proteins	He et al., 2021	[62]
Aging	Promotes weight loss, improves glucose tolerance in high fat diet, accelerates osteopenia and lipopenia	Hsu et al., 2018	[63]
Obesity	Reduces body weight and hyperlipidemia, reduces TNF alpha and IL1, stimulates adiponectin increasing fat burn	Lee et al., 2017	[54]
Alzheimer’s disease	Inhibits learning and memory impairment	Choi et al., 2022	[52]
Ischemic injury	Decreases cell death and pro-inflammatory cytokines	Ryu et al., 2022	[48]
Acute colitis	Anti-inflammatory, anti-apoptotic, protect tight junctions	Nishikawa et al., 2021	[50]
Bone Marrow Stem Cell Extract			
Myocardial infarction	Reduces infarct size, increases vascularity, reduces apoptosis, enhances cardiac function	Yeghiazarians et al., 2009	[64]
Acetaminophen induced liver failure	Reduces necrosis, increases mitotically active cells	Khubutiya et al., 2015	[55]
Osteoradionecrosis	Intravenous injections improve bone recovery	Michel et al., 2017	[51]
NOD mice	Increases salivary gland function proteins, decreases pro-inflammatory markers	Misuno et al., 2014; Ghada et al. 2019	[56,65]
Radiation injury	Increases salivary gland functional proteins, increases angiogenesis, and reparative proteins	Tran et al., 2013; Fang et al., 2015; Su et al., 2018	[42,43,66,67]
Splenic nerve injury	Partially restores serum proteomic homeostasis, reduces pain	Zhou et al., 2022	[68]
Embryonic Stem Cell Extracts			
Cellular reprogramming	Transient colony formation and multi-differential potential seen in rabbit corneal cells	Zhan et al., 2010	[69]
Cellular reprogramming	Transient colony formation and multi-differential potential seen in human granulosa cells	Talaei-Khozani., 2012	[70]
Bone regeneration	Umbilical cord CE cytokines enhance bone regeneration	Byeon et al., 2010	[71]
Other Adult Stem Cell Extracts			
Radiation injury	Increased salivary flow (50–60%) after treatment in vivo	Su et al., 2020	[35]
Mononuclear cells extract for MI	Improves cardiac function, decreases infarct size	Angeli et al., 2012	[47]
Cancer model using axolotl oocyte extract	Causes cell cycle arrest, reduces metabolism and oncogenic signaling, reduces angiogenesis in vitro and in vivo	Saad et al., 2018	[72]
Leukemia using axolotl extract	Cell cycle arrest in human acute myeloid leukemia HL-60 cell line	Suleiman et al., 2020	[49]
Ultraviolet radiation protection with Crocodile WBC	Promotes cell proliferation, reduces UV-induced morphological changes, reduces pigmentation	Joompang et al., 2022	[73]
Cancer model with Crocodile WBC	Decreases the mitochondrial membrane potential of HeLa cells, induces apoptotic death	Patathananone et. al., 2016	[74]
Tissue Extracts			
Wound healing using ADSC tissue extract	Promotes keratinocyte proliferation and stimulates fibroblast and adipose stem cells migration in vitro	Lopez et al., 2018	[75]
Wound healing using ADSC tissue extract	Increases vessel density and formation of neo adipocytes in vivo, promotes the tube formation of human HUVECs	He et al., 2019	[76]
Wound healing using ADSC tissue extract	Increases the proliferation and migration of dermal fibroblasts, increases the thickness of the dermis	Xu et al., 2020	[77]

**Table 2 biology-12-00305-t002:** Effects of dosage and frequency of cell extracts.

Condition	Stem Cell Type	Dosage/Frequency	Outcome	Ref.
Wound healing in murine model	ADSC	200 mg/200 mL extract from 4 × 10^7^ cells, every two days after injury	Smaller wound area, increased fibroblast migration	[46]
Wound healing in murine model	ADSC	60 mL extract obtained from 1 × 10^6^ cells, injected intradermally around the wound at four sites and 40 mL onto the wound bed, once	ADSC extract treatment showed less re-epithelization and healing than cell only group	[60]
Obesity in murine high fat diet model	ADSC	50 mL, lysate from 71,428 cells/kg body weight, daily from 4-week age for 10 weeks	Reduced body weight and lipidemia in treated groups	[54]
Ageing in mice	ADSC	3 times a week, every second month, starting at 12 months of age until natural death (3-year life-long experiment)	Shortened average life span, greater bone loss and increased lean mass	[63]
Radiation injury in mice models	BMMSC	100 μL of extract derived from 10^7^ cells/100 μL, treatment started: 1,3- and 7-weeks post radiation; frequency: 1, 2, 3 and 5 weekly injections	Treatment within 3 weeks and 5 weekly injections showed most favorable results	[83]
Acute colitis	ADSC	0.2 mL extract per mice, derived from 1 × 10^6^ cells/200 μL, day 2, 3, and 4 after colitis initiation	Reduced disease activity index score with multiple injections versus single.	[50]

**Table 3 biology-12-00305-t003:** Routes of administration of stem cells and their products to SGs.

Advantages	Disadvantages
1. Intraglandular	
- Safe and effective - Ultrasound-assisted techniques allow precise implantation.	- Invasive
2. Ductal cannula instillation	
- Safe and effective - Targeted approach - Non-invasive	- Unfeasible in some cases
3. Intraperitoneal	
- Not technique sensitive - More systemic distribution, bypassing lung	- Invasive - Poor cell engraftment - Requires a higher dosage
4. Intra-venous	
- Safe - Least invasive - Most common - More systemic distribution	- Poor cell engraftment, more lung, and liver distribution - Requires a higher dosage

**Table 4 biology-12-00305-t004:** Methods for preparing mammalian cell extracts (for a more detailed methodology, readers should refer to Islam et al., 2017) [88].

Method Category	Cell Sources	Advantages	Disadvantages	References
Ultrasonication method	ADSC, WBC	- Quick and efficient - Independent of cell type	- Degradation of enzymes is often observed - Large amount of heat generated which needs to be dissipated - High power requirement	[54,74]
Chemical lysis + ultrasonication method	ESC, 3T3 cell line	- Chemical molecules permeate the cells to facilitate their disruption - Suitable for sensitive proteins and all cell types - As chemical disruption can cause incomplete cell lysis, combination with sonication can aid the disruption process	- The chemicals can be potentially toxic and cause immune reaction - Time consuming	[69,70]
Osmotic+ ultrasonication method	ADSC	- Osmotic methods is useful for eukaryotic cells due to their fragile plasma membrane - Can be used for sensitive intracellular products	- Osmosis is not useful for all cell types (prokaryotes with cell walls)	[52,62]
Temperature treatment	ADSC, UCB, BMC, BMMSC, MC, GC, RBC, LSC	- Safe as no chemical substance used - Most efficient for extracting highly expressed proteins - Isotonic with physiologic pH	- Time consuming - Can cause damage to intracellular proteins and components	[60,65,66,68,71,67,83,34,35]
Osmotic + temperature treatment	ADSC	- Combine the efficiency of both disruption methods - Can overcome the limitation of temperature treatment to protect sensitive proteins	-	[61,63]

**Table 5 biology-12-00305-t005:** Comparison of cell extract therapy to other stem cell therapies.

Cell Extract (CE)	Advantages	Limitations
CE vs. Cell-Based Therapies	- Smaller immune reaction, both autologous and allogenic grafts possible - Less cell culture time - Limited chance for tumor formation and malignancy - Easier storage and transport - More feasible for clinical application	- Composition, active factors, and mechanism of action of the extract remain unknown - Impact of age and the donor condition can lead to variation in composition and effectiveness - Standard isolation and derivation procedures required for consistency and reproducibility - Multiple injections are required, which can reduce patient compliance - Long-term effects need to be studied to eliminate the risk of cancer recurrence and other side effects, if any
CE vs. Other Cell-Free Therapies	- Mechanical preparation ensures a more natural composition without any chemical contamination - Cost efficient as longer growth and maintenance of cells is not required - No additional supporting cell treatment is required - High yield per cell, thus a smaller number of cells required - Less time requirement ensures better feasibility	

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
