# Peer review of "Cell-Free Therapies: The Use of Cell Extracts to Mitigate Irradiation-Injured Salivary Glands"

_biology, 2023, doi:10.3390/biology12020305_

Round 1

Reviewer 1 Report

In this review, the authors do a comprehensive study of cell-free therapies for their use to mitigate irradiation-injured salivary glands. The manuscript is well written, clear and will be of interest for a broad audience. I would only suggest some changes/minor points:

1. Although it is described in the abstract, I would use the term ‘IR-induced SG’ throughout the manuscript instead of IR-SG.

2. Could the authors include a Table for section 3.1 similar to the other tables used in the manuscript?

3. In section 3.3, lines 375 and 379, the right table is Table 3. Although I think that the reference in line 379 is unnecessary.

4. In Table 3, could the authors add an extra column showing the advantages and disadvantages of each method?

5. In section 4, in the first paragraph I would also mention the importance of making comparisons between different donors.

6. In section 4, if posible I would add a Figure/Table showing the studies and results of stem cell extracts profiles. 

Author Response

In this review, the authors do a comprehensive study of cell-free therapies for their use to mitigate irradiation-injured salivary glands. The manuscript is well written, clear and will be of interest for a broad audience. I would only suggest some changes/minor points:

RESPONSE: We appreciate the time and effort you have dedicated to providing insightful comments on ways to strengthen our manuscript.

1.Although it is described in the abstract, I would use the term ‘IR-induced SG’ throughout the manuscript instead of IR-SG.

RESPONSE: We are grateful to the Reviewer for the comment. To make the manuscript clearer, we have now revised all “IR-SG” to “IR-induced SG.”

  1. Could the authors include a Table for section 3.1 similar to the other tables used in the manuscript?

RESPONSE: Thank you for your suggestion. We have now added another table, ‘Table 2:  Effects of dosage and frequency of cell extracts’, to the manuscript under the aforementioned section.

  1. In section 3.3, lines 375 and 379, the right table is Table 3. Although I think that the reference in line 379 is unnecessary.

RESPONSE: Thank you for your comment on improving our manuscript, and we are sorry for the mistake. We deleted all the references from lines 375 to 379 since they had been listed in table 3 below.

  1. In Table 3, could the authors add an extra column showing the advantages and disadvantages of each method?

RESPONSE: Thank you for this constructive suggestion. We have now added the columns (Advantages and disadvantages) to the table ‘Table 4. Methods for the preparation of mammalian cell extracts.

  1. In section 4, in the first paragraph, I would also mention the importance of making comparisons between different donors.

RESPONSE: We thank the reviewer’s comment. We have added this info to the last paragraph in section 4 (Lines 1179-1184).

Lastly, individual differences, including age, gender, and physical condition, might influence the cell extracts' constituents and treatment effects. Therefore, it is important to clarify the difference by comparing the CEs from many donors.  And individual differences should be considered by the researchers when preparing the CEs from human cells or tissues and utilizing them in the clinic.

  1. In section 4, if possible, I would add a Figure/Table showing the studies and results of stem cell extracts profiles. 

RESPONSE: We have now added the figure, ‘Figure 1: Proteomic profiling of ADSC using Mass spectroscopy by He et al., 2017’.

Overall, we have extensively revised the manuscript to correct any grammatical errors.

Reviewer 2 Report

In the manuscript the authors summarize cell-free therapies for the treatment of IR-induced salivary gland (SG) hypofunction (IR-SG), focusing in particular on the administrations of diverse cell extract. They compare and discuss the efficacy of cell extracts derived from different sources and obtained with different procedures. They also compare the efficacy and vantages of the existing cell therapies and cell-free therapies.

The manuscript is interesting, well organized and clearly described.

Minor suggestions

“Paragraph 5.1 Cell extract and cell-based therapies”.

The description of cell therapies for the treatment of IR-SG is poorly described and should be more elaborate

Line 591. Two references should be added :

 n 38  Sumita, Y.; Liu, Y.; Khalili, S.; Maria, O.M.; Xia, D.; Key, S.; Cotrim, A.P.; Mezey, E.; Tran, S.D. Bone marrow-derived cells 956 rescue salivary gland function in mice with head and neck irradiation. Int J Biochem Cell Biol 2011, 43, 80-87, doi:10.1016/j.bi-957 ocel.2010.09.023

- Lim JY, Ra JC, Shin IS, Jang YH, An HY, Choi JS, Kim WC, Kim YM. Systemic transplantation of human adipose tissue-derived mesenchymal stem cells for the regeneration of irradiation-induced salivary gland damage. PLoS One. 2013 Aug 9;8(8):e71167. doi: 10.1371/journal.pone.0071167. PMID: 23951100; PMCID: PMC3739795.

Author Response

In the manuscript the authors summarize cell-free therapies for the treatment of IR-induced salivary gland (SG) hypofunction (IR-SG), focusing in particular on the administrations of diverse cell extract. They compare and discuss the efficacy of cell extracts derived from different sources and obtained with different procedures. They also compare the efficacy and vantages of the existing cell therapies and cell-free therapies. The manuscript is interesting, well organized and clearly described.

RESPONSE: We appreciate the time and effort you have dedicated to providing insightful comments on ways to strengthen our manuscript.

Minor suggestions

1 “Paragraph 5.1 Cell extract and cell-based therapies”.

The description of cell therapies for the treatment of IR-SG is poorly described and should be more elaborate.

RESPONSE: Thank you for your comment on improving our manuscript. We have added more description of cell-based therapies for IR-SG in section 5 (Lines 1200-1342).

Numerous cell-based therapies have been reported as potential regenerative therapies to relieve xerostomia caused by radiotherapy [33, 36, 37, 41, 49, 117, 118]. Woodward et al. demonstrated that bone marrow stem cells secreted growth factors, mitigated the immune response, alleviated inflammation, and promoted the remaining local stem cell to differentiate and proliferate in SGs [119]. Other adult tissue-specific stem cells, such as adipose stem cells and SG stem cells, also showed potential in treating salivary hypofunction. Three studies have revealed that adipose-derived stem cells alleviated the hypofunction of the salivary gland post-irradiation in mice [118, 120] and rat models [121]. Recently, adipose stem cells have been tested in phase I–II clinical trials [40]. The other candidate cell source for treating irradiation-induced hyposalivation is the stem/progenitor cells in salivary glands. Dr. Coppes group isolated and cultured the murine SG cells into the spheres and reported that the cells were positive to stem cell markers, such as CD117, CD24, CD29, CD49f, CD44, CD90, CD34, Sca-1, Mushashi-1, and c-Kit [122]. Then, the human SGs-derived spheres were successfully formed with the cells positive to c-Kit [123-125]. These c-Kit positive stem cells could self-renew for more than 48 weeks in vitro and in vivo [122, 123, 126, 127]. Furthermore, the c-Kit+ cells rescued hyposalivation and maintained the homeostasis of SGs post-irradiation [122, 127]. More recently, the studies about SG c-Kit+ cells were tested in phase I–II clinical trials [40, 123, 125, 128]. In addition to the c-Kit+ cells, there is another tissue specific-stromal/ mesenchymal stem cell isolated from human major SGs [129, 130], capable of multiple differentiation, including osteogenesis, adipogenesis, chondrogenesis and could generate epithelial cell type (epithelial and hepatic cells). Furthermore, these stem cells showed capabilities in treating SG hypofunction induced by irradiation. Apart from the stem cells, several other cell sources have been studied in SG regeneration, such as bone marrow cells [38], peripheral blood mononuclear cells [39], and dental pulp cells [36]. Additionally, there are currently other potential cell sources, such as Wharton jelly stem cells [131], menstrual blood-derived cells [132, 133], and bone marrow-derived mononuclear cells[134-136], which exhibit their potential in treating many other diseases. However, limited evidence shows their application in SG repair and regeneration.

2 Line 591. Two references should be added:

 Sumita, Y.; Liu, Y.; Khalili, S.; Maria, O.M.; Xia, D.; Key, S.; Cotrim, A.P.; Mezey, E.; Tran, S.D. Bone marrow-derived cells rescue salivary gland function in mice with head and neck irradiation. Int J Biochem Cell Biol 201143, 80-87, doi:10.1016/j.bi-957 ocel.2010.09.023

- Lim JY, Ra JC, Shin IS, Jang YH, An HY, Choi JS, Kim WC, Kim YM. Systemic transplantation of human adipose tissue-derived mesenchymal stem cells for the regeneration of irradiation-induced salivary gland damage. PLoS One. 2013 Aug 9;8(8):e71167. doi: 10.1371/journal.pone.0071167. PMID: 23951100; PMCID: PMC3739795.

RESPONSE: We are grateful to the Reviewer for the comment. We have now added these references in section 5.1(Lines 1195-1197).

Numerous cell-based therapies have been reported as potential regenerative therapies to relieve xerostomia caused by radiotherapy [33, 36, 37, 41, 49, 117, 118].

Overall, we have extensively revised the manuscript to correct any grammatical errors.

Reviewer 3 Report

The authors provide a comprehensive review of the use of different types of mesenchymal stem cells from different sources. It is an orderly study and details each of the aspects related to the use of this type of therapy for the treatment of salivary gland hypofunction after radiotherapy treatment in the case of head and neck cancer. They list both strengths and weaknesses of each of the stem cell treatments or cell-free treatments where their experiments are mainly focused on obtaining cell extracts from different sources of stem cells for the treatment of different diseases, focusing their work on obtaining stem cells from adipose tissue (ADSCE). The main focus is on cell-free therapy as an alternative strategy to conventional treatments. Not only do they review the ability to regenerate the salivary glands' ability to produce a normal amount of saliva and to produce saliva of good quality, but they also explain that this therapy has the ability to regenerate the damaged cells themselves after irradiation, whether they are salivary glands, endothelial or epithelial cells. The whole manuscript is based on the fact that mesenchymal stem cell therapy exerts its main mechanism through a paracrine action and that is why they make this literature review of this therapy based on mesenchymal stem cell extracts as a therapy with numerous advantages because it is safer, more practical and more economical compared to cell-based therapy.

The concept of this manuscript is interesting but the author should consider some suggestions:

-Page 3 line 119. I don't understand why ‘cell extract’ is written in inverted commas.

-The authors indicate that the best results are obtained from stem cells obtained from adipose tissue. It has been described that these cells are more mature and differentiated than bone marrow-derived stem cells. This may be the reason why stem cells obtained from different sources have different effects in the treatment of diseases. There are currently other sources of stem cells that the authors do not list, such as those obtained from Warton's jelly, cells obtained from menstrual blood, etc. The authors should consider this observation in the manuscript.

- As the author describes in the text, BMCE can have pro-inflammatory effects because of the type of cells that appear in the bone marrow, such as granulocytes. Therefore, the corresponding paragraph between lines 267-270 should be explained by concluding the negative effect that BMCE can have.

- As for obtaining cell extracts, the authors state that the best method is freeze-thawing of the cells. They explain that it is a safer and more stable method than other methods because no chemical reagents are used. Have the authors found that this process ensures the structural stability of the proteins necessary for this therapy, and can these proteins degrade after freezing-thawing? The authors point out the importance of working with the native protein without any degradation process. The authors should make this clear.

- Why do the authors say that cell extracts show less immune response than mesenchymal stem cells? It is clear that the proteins in the cell extracts can activate the immune system cells even more strongly than the stem cell itself. In addition, mesenchymal stem cells have been described as having other benefits that have not been described in cell extracts, for example the capacity for immunomodulation as they can regulate the phenotype and type of immune cell that is attracted to the damaged tissue. This needs to be explained by the authors.

-References are incomplete, only up to reference 113 when in the text there are up to 170. This problem should be fixed or corrected.

Author Response

The authors provide a comprehensive review of the use of different types of mesenchymal stem cells from different sources. It is an orderly study and details each of the aspects related to the use of this type of therapy for the treatment of salivary gland hypofunction after radiotherapy treatment in the case of head and neck cancer. They list both strengths and weaknesses of each of the stem cell treatments or cell-free treatments where their experiments are mainly focused on obtaining cell extracts from different sources of stem cells for the treatment of different diseases, focusing their work on obtaining stem cells from adipose tissue (ADSCE). The main focus is on cell-free therapy as an alternative strategy to conventional treatments. Not only do they review the ability to regenerate the salivary glands' ability to produce a normal amount of saliva and to produce saliva of good quality, but they also explain that this therapy has the ability to regenerate the damaged cells themselves after irradiation, whether they are salivary glands, endothelial or epithelial cells. The whole manuscript is based on the fact that mesenchymal stem cell therapy exerts its main mechanism through a paracrine action and that is why they make this literature review of this therapy based on mesenchymal stem cell extracts as a therapy with numerous advantages because it is safer, more practical and more economical compared to cell-based therapy.

The concept of this manuscript is interesting but the author should consider some suggestions:

1 -Page 3 line 119. I don't understand why ‘cell extract’ is written in inverted commas.

RESPONSE:  We have now removed the inverted commas. We initially used them to highlight the term cell extract for the readers. However, based on the reviewer’s comment, this may be distracting or confusing for the readers; thus, we have removed these inverted commas.

2 -The authors indicate that the best results are obtained from stem cells obtained from adipose tissue. It has been described that these cells are more mature and differentiated than bone marrow-derived stem cells. This may be the reason why stem cells obtained from different sources have different effects in the treatment of diseases. There are currently other sources of stem cells that the authors do not list, such as those obtained from Warton's jelly, cells obtained from menstrual blood, etc. The authors should consider this observation in the manuscript.

RESPONSE:  We are grateful to the Reviewer for the comment. We stated in the manuscript that adipose stem cell is one of the most promising adult stem cells for clinical application with various advantages. But there is no research comparing the effect of different stem cells from different sources in treating salivary gland diseases. So, we can’t conclude that adipose stem cells provided the best result compared to other stem cells, such as bone marrow stem cells.

Indeed, we should have listed many other sources of stem cells in the manuscript. Because we are focusing on the effect of cell extract on IR-induced SG damage, we have limited our discussion to this condition and the cells utilized in the literature so far. But as the reviewer suggested, it is necessary to discuss the cell base therapy of IR-SG more. Therefore, we have added a new paragraph describing the treatment of different stem cells used to restore the function SG (Lines 1200-1342).

Other adult tissue-specific stem cells, such as adipose stem cells and SG stem cells, also showed potential in treating salivary hypofunction. Three studies have revealed that adipose-derived stem cells alleviated the hypofunction of the salivary gland post-irradiation in mice [118, 120] and rat models [121]. Recently, adipose stem cells have been tested in phase I–II clinical trials [40]. The other candidate cell source for treating irradiation-induced hyposalivation is the stem/progenitor cells in salivary glands. Dr. Coppes group isolated and cultured the murine SG cells into the spheres and reported that the cells were positive to stem cell markers, such as CD117, CD24, CD29, CD49f, CD44, CD90, CD34, Sca-1, Mushashi-1, and c-Kit [122]. Then, the human SGs-derived spheres were successfully formed with the cells positive to c-Kit [123-125]. These c-Kit positive stem cells could self-renew for more than 48 weeks in vitro and in vivo [122, 123, 126, 127]. Furthermore, the c-Kit+ cells rescued hyposalivation and maintained the homeostasis of SGs post-irradiation [122, 127]. More recently, the studies about SG c-Kit+ cells were tested in phase I–II clinical trials [40, 123, 125, 128]. In addition to the c-Kit+ cells, there is another tissue specific-stromal/ mesenchymal stem cell isolated from human major SGs [129, 130], capable of multiple differentiation, including osteogenesis, adipogenesis, chondrogenesis and could generate epithelial cell type (epithelial and hepatic cells). Furthermore, these stem cells showed capabilities in treating SG hypofunction induced by irradiation. Apart from the stem cells, several other cell sources have been studied in SG regeneration, such as bone marrow cells [38], peripheral blood mononuclear cells [39], and dental pulp cells [36]. Additionally, there are currently other potential cell sources, such as Wharton jelly stem cells [131], menstrual blood-derived cells [132, 133], and bone marrow-derived mononuclear cells [134-136],  which exhibit their potential in treating many other diseases. However, limited evidence shows their application in SG repair and regeneration.

3 - As the author describes in the text, BMCE can have pro-inflammatory effects because of the type of cells that appear in the bone marrow, such as granulocytes. Therefore, the corresponding paragraph between lines 267-270 should be explained by concluding the negative effect that BMCE can have.

RESPONSE:  We are sorry for this misunderstanding. The BMCE did not induce any obvious pro-inflammatory effect, while one of its cell fraction (the GCE) did [1]. It is because the percentage of granulocytes in BMCE is very low and would not induce a significant immune response in the mouse model. Therefore, the pro-inflammatory effect is not a disadvantage of the BMCE treatment, but only if we isolate one of its cell fraction (GCE) and gives it as a whole [1]. We have now added this explanation in section 2.1(Lines 225-228, 318-320).

We have also edited the summary as: “In summary, several cell extracts have demonstrated their effectiveness in treating SG hypofunction, including ADSCE, BMSCE, BMCE, MCE, LSCE, and spleen cell extracts. GCE was effective but also induced an acute inflammatory response. The optimal cell source is still unknown, and further experiments are needed to address this issue.”

“Results showed that BMCE and MCE provided therapeutic efficacy by improving the secretory function of IR-injured SG. Both of these cell extracts did not induce an obvious immune response; GCE was of shorter efficacy but induced an acute inflammatory response.”

4 - As for obtaining cell extracts, the authors state that the best method is freeze-thawing of the cells. They explain that it is a safer and more stable method than other methods because no chemical reagents are used. Have the authors found that this process ensures the structural stability of the proteins necessary for this therapy, and can these proteins degrade after freezing thawing? The authors point out the importance of working with the native protein without any degradation process. The authors should make this clear.

RESPONSE:  Thank you for your valuable comment. We did not assume that the freeze-thaw method could protect all the proteins without degradation. Indeed, ensuring that no protein degrades during a series of isolation processes is difficult (We have now added the advantages and disadvantages in the table ‘Table 4. Methods for the preparation of mammalian cell extracts’). Several specific proteins may degrade after freeze and thaw cycles, but they would also degrade or even degenerate after other preparation methods, such as ultrasonication. But the obvious therapeutic effect of cell extracts prepared by freeze and thaw cycles has been reported by many studies (listed in the manuscript). It indicates that although several proteins may degrade during the preparation process, these degraded proteins might not be as crucial. Lastly, we have mentioned that the freeze-thaw method is the most common method used in studies and has several advantages compared with other methods. But we still need to state that it is not the ideal method for cell extract preparation, and optimization is still needed to improve this method. We concluded in the first sentence of the last paragraph in section 3 that hat “In conclusion, it is necessary to compare the constituent and the effect of the cell extracts prepared by different methods and optimize these combination techniques to increase the efficacy and stability for both disease treatment and the isolation methods of cell extracts.”

5 - Why do the authors say that cell extracts show less immune response than mesenchymal stem cells? It is clear that the proteins in the cell extracts can activate the immune system cells even more strongly than the stem cell itself. In addition, mesenchymal stem cells have been described as having other benefits that have not been described in cell extracts, for example the capacity for immunomodulation as they can regulate the phenotype and type of immune cell that is attracted to the damaged tissue. This needs to be explained by the authors.

RESPONSE: Thank you for your valuable comment. We state that the cell extract shows less immune response than intact cells (not specific to MSC). It is because cell extract contains fewer histocompatibility antigens, such as MHC-I and MHC-II, than the intact cells and should theoretically elicit a weaker immune response [2, 3]. One study injected human BMCE into the mouse model and reported the significant treatment effect without immune rejections[4].

As the reviewer mentioned, MSC shows the capacity of immunomodulation. Evidence suggests that the paracrine effect is the primary mechanism for cell-based therapy [5-7]. The paracrine factor released from cells allowed tissue repair and regeneration via modulating the immune reaction, mitigating inflammation and fibrotic effects, promoting angiogenesis and neurogenesis, and preventing apoptosis [2, 6, 8, 9]. The MSC paracrine factors involved in immunomodulation, such as SDF-1, can also be detected in several cell extracts [1, 2, 10, 11].  It indicated that cell extract could regulate the immune response as stem cells do (Lines 994-997).

Stromal cell-derived factor-1 (SDF-1) is another critical factor in CE that plays a role in disease treatment [104]. It shows the potent capacity to repair damaged tissues by regulating immune response, inflammation, cell migration, vascularization, and neurogenesis [104-106].

6 -References are incomplete, only up to reference 113 when in the text there are up to 170. This problem should be fixed or corrected.

RESPONSE: Thank you for your valuable comment. We have now corrected all the references in the manuscript.